**Subject Area:**
neuroscience/cellular biology

biomarkers, neurodegeneration, exosomes, miRNA, ALS/MND

**Author for correspondence:**
Paul Alan Cox
e-mail: paul@ethnomedicine.org

# An miRNA fingerprint using neural-enriched extracellular vesicles from blood plasma: towards a biomarker for amyotrophic lateral sclerosis/motor neuron disease

Sandra Anne Banack, Rachael Anne Dunlop and Paul Alan Cox

Brain Chemistry Labs, Institute for Ethnomedicine, PO Box 3464, Jackson, WY 83001, USA

PAC, 0000-0001-6401-2981

Biomarkers for amyotrophic lateral sclerosis/motor neuron disease (ALS/MND) are currently not clinically available for disease diagnosis or analysis of disease progression. If identified, biomarkers could improve patient outcomes by enabling early intervention and assist in the determination of treatment efficacy. We hypothesized that neural-enriched extracellular vesicles could provide microRNA (miRNA) fingerprints with unequivocal signatures of neurodegeneration. Using blood plasma from ALS/MND patients and controls, we extracted neural-enriched extracellular vesicle fractions and conducted next-generation sequencing and qPCR of miRNA components of the transcriptome. We here report eight miRNA sequences which significantly distinguish ALS/MND patients from controls in a replicated experiment using a second cohort of patients and controls. miRNA sequences from patient blood samples using neural-enriched extracellular vesicles may yield unique insights into mechanisms of neurodegeneration and assist in early diagnosis of ALS/MND.

## 1. Background

Neurodegenerative diseases such as Alzheimer's disease (AD), Parkinson's disease (PD) and amyotrophic lateral sclerosis/motor neuron disease (ALS/MND) continue to provide challenges for diagnosis since validated, clinically useful diagnostic biomarkers are currently unavailable. Rapid diagnosis and intervention in the disease process could be beneficial in slowing neurodegenerative disease progression as well as facilitating the testing of new therapies. In ALS/MND, the average time from diagnosis to death is typically short (2–5 years) and it is not unusual for patients to wait a year before receiving a diagnosis [1]. Since disease progression correlates with motor neuron loss, early intervention could be critical for the development of new effective drug therapies. Current investigative drugs suggest some hope to reduce the rate of ALS/MND disease progression [2]; however, the discovery of biomarkers would be a tremendous asset to these efforts.

To date, an ALS/MND diagnosis is based on clinical features with the elimination of alternative diagnoses and supporting data retrieved from electromyograms, nerve conduction studies, muscle biopsies, magnetic field imaging and biofluid analysis [3,4]. The search for ALS/MND biomarkers useful for diagnosis, prognosis and analysis of drug efficacy includes a variety of molecules found in biofluids and other techniques including: heavy and light chain neurofilaments, TAR DNA-binding protein 43 (TDP-43), a lipid peroxidation product (4-hydroxy-2,3-nonenal), a urinary neurotrophin receptor p75 extracellular domain, cystatin C, mRNA, miRNA, extracellular glutamate, markers

of inflammation, microglial activation, electrical impedance myography, rate of disease progression, spinal cord imaging and others [1,5–17]. Thus far, none of these biomarkers have been sufficiently validated to be incorporated into the clinical standard of care [1,3,9,15,18].

Biomarker exploration has increased in recent years due to advances in cellular biology. The discovery of intercellular communication through exosomes has initiated new avenues for biomarker exploration. Exosomes are characterized as lipid membrane vesicles of endosomal origin of 30–200 nm in size, that contain a heterogeneous mix of messenger RNA (mRNA), microRNA (miRNA), transfer RNA (tRNA), Y RNA, small non-coding RNA (sRNA), DNA, lipids and proteins [19]. Extracellular vesicles (EVs) are a more inclusive term for nucleus-absent, lipid bilayer particles, including exosomes, that are naturally released from the cell [20]. EVs released into the extracellular matrix and taken up by adjacent cells impact cellular function of the recipient cells and possess both therapeutic and pathogenic potential [21]. EVs are thought to be expelled from all cell types and can be isolated from diverse biological fluids including cerebrospinal fluid (CSF), plasma, serum, breast milk, lymph, bile and saliva. EVs are remarkably stable in bodily fluids, providing protection for their molecular cargo from enzymatic breakdown. This stability combined with their availability in easily obtainable biological fluids make them of interest as potential reservoirs for disease biomarkers, which in turn could be useful for assessing the efficacy of therapeutic interventions [22]. Cancer research has demonstrated that changes in disease progression correlate with biological changes within EVs [23,24] and this has made EV analysis a particularly attractive option for biomarker research.

In parallel with current research on EVs, the investigation of miRNA has independently shown promise in the quest for biomarkers. Since miRNA are post-transcriptional regulators of gene expression, mediated via suppression of the translation of mRNAs or degradation of target mRNAs, they transmit executable instructions. They have been identified as potential biomarkers in many fields including cancer [25], AD [26], systemic lupus erythematosus [27], traumatic brain injury [28], cardiovascular disease [29], PD [30], multiple sclerosis [31] and diabetes [32]. Since miRNA are found as cargo within EVs and the lipid membrane surrounding EVs protects the miRNA from enzymatic degradation, there is good rationale for examining miRNA extracted from isolated EVs. Added to this the potential for selectively enriching EVs by subtypes based on the specific protein surface markers, these techniques can be targeted and potentially produce reliable, stable disease markers.

In this study, we have identified eight miRNA sequences from enriched EV extractions of blood plasma that consistently and significantly differentiate ALS/MND patients from healthy controls. Since the composition of blood extractions probably includes small molecules and some other EV subtypes, we choose here to use the more generic term of EV [20]. Exploiting cell-specific protein markers, we isolated a neural-enriched sub-population of extracellular vesicles (NEE) as a mechanism for analysing neural-specific cargo. This technique generates a pool of NEE that have the potential to be much more specific, reliable, and repeatable than other sources of biomarkers [33]. We compared the miRNA cargo from NEE in order to examine differential expression of miRNA between plasma samples from healthy controls and ALS/MND patients. A complete replication of these biomarkers using identical techniques, but a second cohort of individuals supported the usefulness of miRNA fingerprints for further discovery.

# 2. Methods

## 2.1. Clinical samples

Forty total plasma samples were analysed in two independent experiments performed using identical criteria. Ten plasma samples were obtained from a blood draw of 10 ALS/MND patients at the time they enrolled in a Phase IIa human clinical trial (NCT03580616). ALS/MND patients were compared with 10 healthy control plasma samples (Innovative Research Inc., Novi, MI, USA). Following this experiment, a second cohort of 10 ALS/MND patients and 10 controls were independently analysed using the same methods and the results compared for repeatability. ALS/MND patients met the following criteria: (1) diagnosis of probable or definite ALS/MND based on the El Escorial criteria [34] within the last 3 years prior to study enrolment; (2) ALSFRS-R score > 25 and a FVC score ≥ 60% predicted; (3) age ≥ 18 years old. Prescription medications of both Riluzole and Endaravone/Radicava were allowed as long as the patient had taken these FDA-approved drugs for three months prior to trial enrolment and maintained a stable dose throughout the trial. None of the ALS/MND patients had a diagnosis or previous history of ischemic stroke, brain tumour, uncontrolled diabetes, renal insufficiency or severe hypertension. Severe hypertension (asymptomatic or hypertensive urgency) was defined as severely elevated blood pressure (180 mm Hg or more systolic, or 110 mm Hg or more diastolic) without acute target organ injury. None of the ALS/MND patients had a diagnosis or previous history of peripheral neuropathy or any other comorbid progressive neurodegenerative disease such as AD, PD, Lewy body disease, Pick's disease, Huntington's disease or progressive supranuclear palsy. None of the ALS/MND patients were undergoing any chemotherapy or radiation therapy for any cancer. None were pregnant women or women who were breast feeding a child. Genetic analysis was not performed on these patients as it was outside the scope of this study. In addition to the 20 control patients identified above, an additional four healthy control plasma samples were used in a pilot study to determine if there was sufficient material for RNA extraction, NGS and qPCR (Qiagen Genomic Services). The pilot study validation also compared miRNA content of different extraction fractions to examine them for distinct signatures.

## 2.2. Plasma extraction

Venous blood was drawn into K2 EDTA tubes followed by immediate centrifugation at 2000$g$ for 15 min (4°C). The plasma was removed prior to being frozen at −80°C. Time between blood collection and freezing was less than 1 h.

## 2.3. EV extraction

Plasma samples were thawed on ice or at 4°C, treated with thrombin to remove fibrinogen, and the EVs were precipitated using polyethylene glycol (SBI ExoQuick, cat. no. EXOQ5TM-1, System Biosciences, Palo Alto, CA, USA). L1

royalsocietypublishing.org/journal/rsob   Open Biol. **10**: 200116

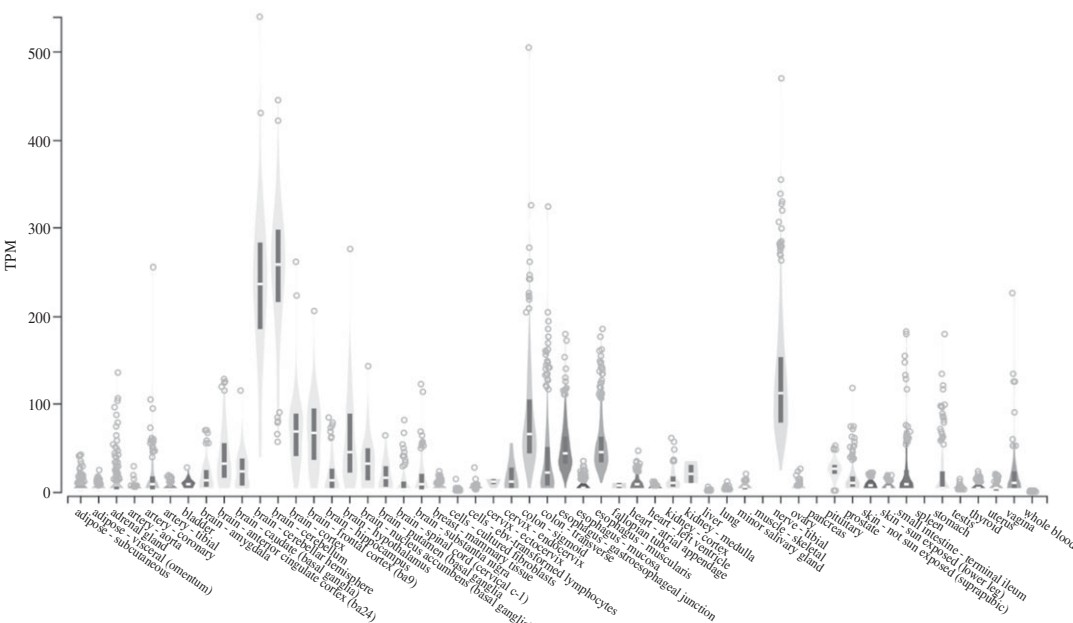

**Figure 1.** L1 cell adhesion molecule (L1CAM) expression is enriched in the brain. Expression values are shown in transcripts per million (TPM). Figure is from the Genotype-Tissue Expression (GTEx) Project which is supported by the Common Fund of the Office of the Director of the National Institutes of Health, and by NCI, NHGRI, NHLBI, NIDA, NIMH and NINDS (GTEx Portal Analysis Release V8, dbGaP Accession phs000424.v8.p2, accessed 22 January 2020, https://gtexportal.org/home/gene/L1CAM); see [35].

cell adhesion molecule (L1CAM) antibodies were used to selectively separate NEE [33]. Since L1CAM is a neural adhesion molecule and is highly expressed in brain and neural tissues (figure 1) [35], this step creates a neural-enriched fraction of EVs with characteristics consistent with exosomes. In brief, 500 µl of plasma was incubated with 15 µl thrombin at room temperature for 30 min. To this 485 µl of sterile Dulbecco's phosphate-buffered saline balanced salt solution (DPBS calcium- and magnesium-free, Caisson Labs PBL01, Smithfield, UT, USA) mixed with three times the recommended concentrations of Halt protease inhibitor cocktail (cat. no. 78429, Thermo Fisher Scientific, Waltham, MA, USA) and Halt phosphatase inhibitor cocktail (cat. no. 78426, Thermo Fisher Scientific, Waltham, MA, USA) was added. The mixture was then centrifuged at $4500g$ for 20 min (4°C). To the supernatant, ExoQuick precipitation solution (252 µl, cat. no. SBI EXOQ20A-1, System Biosciences Inc, Palo Alto, CA, USA) was then added to precipitate extracellular vesicles and the solution was incubated at 4°C for 1 h. The sample was centrifuged at $1500g$ for 20 min (4°C), and the supernatant discarded. The pellet was resuspended in 500 µl of ultra-pure water that contained the 3× protease and phosphatase inhibitors, vortexed gently and then placed on a rotating mixer overnight. This fraction represents the total extracellular vesicle extraction.

## 2.4. Neural-enriched EV extraction

Enrichment of neural-enriched EVs was accomplished by the addition of 4 µg of mouse anti-human CD171 (L1 cell adhesion molecule (L1CAM) neural adhesion protein) monoclonal antibody [cat. no. eBIO5G3 (5G3), (13-1719-82), Biotin, eBioscience™ Antibodies, Thermo Fisher Scientific, Waltham, MA, USA] in 50 µl of 3% bovine serum albumin (BSA) (cat. no. 37525, Block BSA 10× in PBS, Thermo Fisher Scientific, Waltham, MA, USA) for 60 min at 4°C on a rotating mixer.

To this solution, we then added 15 µl of streptavidin-agarose resin (cat. no. 53116, Pierce Streptavidin Plus UltraLink Resin, Thermo Fisher Scientific, Waltham, MA, USA) plus 25 µl of 3% BSA. This mixture was incubated at 4°C on a rotating mixer for 30 min followed by the addition of 4 µl of ultrapure 1 M TRIS–HCl pH 8.0 (cat. no. 15568025, Thermo Fisher Scientific, Waltham, MA, USA) to adjust pH to 7.0. The mixture was then centrifuged at $200g$ for 10 min (4°C). The supernatant fraction represents the total heterogeneous extracellular vesicle population minus the extracellular vesicles with L1CAM neural surface proteins, a fraction which we designate as T-N. The pellet containing the neural-enriched EVs (NEE) was then suspended in 200 µl of 0.1 M glycine-HCl (pH 2.5) and the solution was strongly vortexed and centrifuged at $4500g$ for 5 min (4°C). The supernatant was recovered and neutralized with 15 µl 1 M TRIS–HCl pH 8.0. The T-N and the NEE fractions were tested for protein content using Molecular Probes Quant iT Qubit Protein Assay Kit (cat. no. Q33211, Thermo Fisher Scientific, Waltham, MA, USA) using a Qubit 3 fluorometer (cat. no. Q33216, Invitrogen, Thermo Fisher Scientific, Waltham, MA, USA) and frozen in aliquots (−80°C).

## 2.5. Characterization of extracellular vesicles

EVs were characterized using a ZetaView NTA System (Particle Metrix Inc. Henderson, NV, USA) in both light and fluorescence modes (cat. no. EXONTA110A-1, System Biosciences Inc., Palo Alto, CA, USA). Further characterization of surface proteins was conducted using the following kits according to manufacturer's instructions: human CD81 ELISA Kit (Sandwich ELISA) (cat. no. LS-F55938, LSBio Seattle, WA, USA); human CD63 ELISA Kit (Sandwich ELISA) (cat. no. LS-F7104, LSBio Seattle, WA, USA) and Exo-Check exosome antibody array (Neuro) (cat. no. EXORAY500A-8, System Biosciences Inc., Palo Alto, CA, USA).

## 2.6. RNA extraction from EVs, library construction and next-generation sequencing

RNA was isolated from purified NEE using the ExoRNeasy Serum/Plasma Kit (cat. no. 77023, Qiagen, Hilden, Germany) at Qiagen Genomic Services, Frederick, MD, USA. Briefly, 5 µl total RNA was used to prepare the miRNA NGS libraries using the QIAseq miRNA Library Kit (cat. no. 331505, Qiagen, Hilden, Germany). RNA was ligated using adapters containing unique molecular indices (UMIs), and the RNA was converted to cDNA. Amplification of the cDNA was conducted using PCR (22 cycles) during which time PCR indices were added. Following purification, library preparation QC was conducted using a Bioanalyzer 2100 (Agilent Technologies, Santa Clara, CA, USA). Libraries were pooled in equimolar ratios based on quality of the inserts and the concentration measurements, quantified using qPCR, and sequenced on the NextSeq 500 System (Illumina, San Diego, CA, USA). FASTQ files were prepared and checked following de-multiplexing of raw data (bcl2fastq software, Illumina, San Diego, CA, USA; FastQC). For trimming, adapter and UMI information from raw reads was extracted using Cutadapt v. 1.11 [36]. Adapter sequences were removed and reads collapsed by UMI using a Qiagen in-house script. Reads were mapped using Bowtie2 v. 2.2.2 [37] where the criteria for aligning reads to spike-ins, abundant sequence and miRbase (v20) specified that the reads perfectly match reference sequences. EdgeR [38] was used to calculate differential expression and data normalized using trimmed mean of M-values (TMM) normalization [39]. miRNA was identified by mapping to miRBase (v. 20) [40]. The reliability of the identified miRNAs is noted to increase with the number of identified fragments expressed in tags per million (TPM) [39].

## 2.7. RNA extraction for qPCR quantitation of miRNA

Total RNA was extracted from the samples using ExoRNeasy Serum/Plasma Kit (cat. no. 77023, Qiagen, Hilden, Germany) high-throughput bead-based protocol using the QIAcube Connect (cat. no. 9002864, Qiagen, Hilden, Germany) at Qiagen Genomic Services. For miRNA quantitation, RNA was reverse transcribed to cDNA using the miRCURY locked nucleic acid (LNA) RT Kit (cat. no. 339340, Qiagen, Hilden, Germany). The RNA Spike-In Kit for RT (cat. no. 339390, Qiagen, Hilden, Germany) was applied to measure extraction efficiency and as quality control for RNA isolation and cDNA synthesis. Isolation controls were UniSp100 and UniSp101 in experiment 1 and UniSp2 and UniSp4 in experiment 2, and the cDNA synthesis controls for both experiments were UniSp3 and UniSp6. cDNA was diluted 50× and assayed in 10 µl qPCR reactions using the miRCURY LNA SYBR Green PCR kit (cat. no. 339345, Qiagen, Hilden, Germany; Qiagen Genomic Services, Frederick, Maryland, USA); each miRNA sequence (hsa, *Homo sapiens*) was assayed once by qPCR for miR-23a-3p, miR-30c-5p, miR-103a-3p, miR-191-5p, and miR-451a for experiment 1 and miR-103a-3p, miR-23a-3p, miR-30c-5p, miR-142-3p and miR-451a for experiment 2. miR-103a-3p, miR-23a-3p, and miR-30c-5p are known to be expressed in a majority of sample types at a consistent concentration, and therefore were used to evaluate miRNA content of samples. To assess any contribution to miRNA signal from haemolysis in plasma samples, the ratio of the differential expression of miR-451a (highly expressed in thrombocytes) with miR-23a-3p (which has relatively stable expression in serum and is not affected by haemolysis), was determined. A ratio greater than 7.0 indicates an increased risk of haemolysis. Negative controls excluding template from the reverse transcription reaction was performed and profiled in the same manner as the samples and spike-ins.

## 2.8. cDNA qPCR

qPCR of cDNA generated from miRNA was conducted in accordance with MIQE guidelines [41] at Qiagen Genomic Services. The relative quantitation of 34 target miRNAs selected from the NGS results was determined by qPCR using SYBR Green detection on a LightCycler® 480 Real-Time PCR System (Roche, Basel, Switzerland) in 384 well plates. A positive reaction is detected by accumulation of a fluorescent signal. The cycle threshold (Ct) is defined as the number of cycles required for the fluorescent signal to cross the threshold (i.e. exceed background levels). The amplification curves were analysed using Qiagen software (v. 1.5.1.62 SP3), both for determination of Ct and for specificity, according to melt curve analysis.

## 2.9. Data analysis

The most stably expressed genes were selected as housekeeping genes by NormFinder [42] and a geometric mean was calculated using the top 3 (miR-29b-3p, miR-126-5p and miR-146a-5p). Consideration to use additional house-keeping genes was evaluated following a stepwise inclusion protocol comparing the pairwise variation ($Vn/n + 1$) calculated between two sequential normalization factors [43]. Since the pairwise variation of V3/4 was below 0.05, we did not include a fourth house-keeping gene. All cycle-time expression values were, therefore, normalized to the geometric mean of the three most stable genes and standard equations were used to calculate $\Delta Ct$, $\Delta\Delta Ct$ and $2^{-(\Delta\Delta Ct)}$.

## 2.10. Statistical analysis

Differential expression analysis of the miRNA identified through NGS was performed using EdgeR [38] at Qiagen Genomic services. For normalization, the trimmed mean of M-values method based on log-fold and absolute gene-wise changes in expression levels between samples (TMM normalization) was used. Differential expression analysis was estimated by an exact test assuming a negative binomial distribution step ($p < 0.05$ was determined as statistically significant).

We compared the gene fold expression [$2^{-(\Delta\Delta Ct)}$] median scores for ALS/MND patients and controls in each of the two replicated experiments. Plots of the data distributions did not conform to normal distributions, so we used nonparametric analysis, specifically a two-tailed Mann–Whitney *U* Test, to examine two alternative hypotheses for each of the 34 miRNA sequences of interest:

$H_0$: the miRNA sequences were drawn from the same population, e.g. median values of $2^{-(\Delta\Delta Ct)}$ of the miRNA sequence are the same for ALS/MND patients and controls;

$H_1$: The miRNA sequences were drawn from different populations, e.g. median values of $2^{-(\Delta\Delta Ct)}$ of the miRNA

**Table 1.** Differentially expressed miRNA as determined by qPCR of plasma samples comparing a total of 20 ALS patients and 20 controls reported from two identical, independent experiments (designated 1 and 2) using a different cohort of individuals in each experiment. Statistics were performed using a two-tailed Mann–Whitney $U$-test. Median values refer to fold gene expression $2^{-(\Delta\Delta Ct)}$. Fold regulation was reported in a biologically relevant way and is defined as, fold change in cases where fold change is greater than one, and in cases where fold change is less than one, fold regulation equals negative one divided by fold change. Direction of fold regulation indicates differential expression between ALS patients compared to healthy controls where an upregulation indicates higher median expression in ALS patients and downregulation indicates decreased expression.

| experiment | miRNA ID | significance | $Z$-statistic | median control | median ALS | fold regulation | direction |
|---|---|---|---|---|---|---|---|
| 1 | miR-146a-5p | $p < 0.05$ | −2.02 | 1.00 | 1.21 | 1.2 | upregulated |
| 2 | miR-146a-5p | $p < 0.05$ | −2.44 | 1.03 | 1.43 | 1.4 | upregulated |
| 1 | miR-199a-3p | $p < 0.05$ | −2.44 | 0.97 | 1.28 | 1.4 | upregulated |
| 2 | miR-199a-3p | $p < 0.001$ | −3.38 | 1.08 | 2.86 | 2.7 | upregulated |
| 1 | miR-4454 | $p < 0.05$ | 2.44 | 1.10 | 0.54 | −1.7 | downregulated |
| 2 | miR-4454 | $p < 0.05$ | 2.44 | 0.97 | 0.44 | −1.8 | downregulated |
| 1 | miR-10b-5p | $p < 0.01$ | 2.61 | 1.00 | 0.58 | −2.1 | downregulated |
| 2 | miR-10b-5p | $p < 0.0001$ | 4.09 | 1.27 | 0.19 | −7.0 | downregulated |
| 1 | miR-29b-3p | $p < 0.001$ | 0.63 | 1.00 | 0.53 | −1.7 | downregulated |
| 2 | miR-29b-3p | $p < 0.01$ | 3.27 | 0.98 | 0.61 | −1.7 | downregulated |
| 1 | miR-151a-3p | $p < 0.01$ | −2.88 | 0.97 | 1.49 | 1.5 | upregulated |
| 2 | miR-151a-3p | $p < 0.01$ | −2.98 | 1.22 | 2.62 | 2.2 | upregulated |
| 1 | miR-151a-5p | $p < 0.001$ | −3.83 | 1.07 | 1.38 | 1.4 | upregulated |
| 2 | miR-151a-5p | $p < 0.001$ | −3.59 | 1.04 | 3.92 | 3.2 | upregulated |
| 1 | miR-199a-5p | $p < 0.001$ | −3.49 | 1.14 | 1.91 | 1.9 | upregulated |
| 2 | miR-199a-5p | $p < 0.001$ | −3.59 | 1.10 | 5.16 | 4.2 | upregulated |

sequence are different for ALS/MND patients and controls; with the null hypothesis $H_0$ being rejected at $p < 0.05$.

# 3. Results

In two separate experiments using a different cohort of patients and controls for each experiment, we found eight miRNA sequences that were significantly and consistently different between ALS/MND patients and healthy controls (table 1). Five miRNA sequences were upregulated in ALS/MND patients and three were downregulated (figure 2). The value and interpretation of these results is linked to the implementation of careful experimental quality controls which are reported below.

## 3.1. EV characterization

The absolute purity of the NEE fraction used in this experiment was not known; therefore, we refer to our extraction in the larger sense of extracellular vesicles *sensu* Théry *et al.* [20]. Nevertheless, all indicators suggest that the miRNA expression reflect differences found within neural-enriched exosomes. The particles recovered from our extraction procedures were of an appropriate size and composition to be consistent with exosomes [20,44,45] with a median peak size of 102 nm for NEE (table 2). The tetraspanins CD81 and CD63 which are known to be enriched in many exosomes were abundant and concentrated in NEE (CD81: 5.2–8.7 billion $\mu l^{-1}$, $n = 20$; CD63: 7.4–13.7 billion $\mu l^{-1}$, $n = 6$). We also note that tumour susceptibility gene 101 (TSG101), a component of the endosomal sorting complexes required

for transport (ESCRT-I) complex, which is common in exosomes, was present and that a marker of cell contamination, calnexin, was absent. In the NEE fraction, we also found the presence of neural markers including: L1 transmembrane, neural cell adhesion, total tau, glutamate receptor 1 and proteolipid proteins (figure 3).

## 3.2. Next generation sequence pilot study

The NGS pilot analysis was conducted to examine the quality and quantity of RNA within the NEE fraction, and to determine if the isolated NEE fraction differed from the T-N EV fraction in miRNA content. We successfully prepared, quantified and sequenced miRNA NGS libraries for all samples. The data passed all QC metrics; the NGS data had a high Q-Score (greater than 30), indicating good technical performance of the NGS experiment. An average of 4.6 million unique molecular index (UMI)-corrected reads per sample were obtained and the average percentage of mapable reads was 32.0%. We identified 256 miRNAs with $\geq 1$ tags-per-million mapped reads (TPM) and 149 with $\geq 10$ TPMs. Results comparing UMI corrected reads (3.8 million: 5.3 million), miRNA/smallRNA (6.6% : 4.7%) and mapped genome (23.9% : 23.8%) did not reveal notable differences between the two groups T-N and NEE, respectively ($n = 4$ for each category). NormFinder analysis returned 25 stably expressed miRNAs, including the constitutively expressed miR-103a-3p with abundance measures between 294 and 3966 average TPM. Comparison between T-N and NEE identified 39 differentially expressed miRNA ($p$-values < 0.05), within this small NGS pilot study.

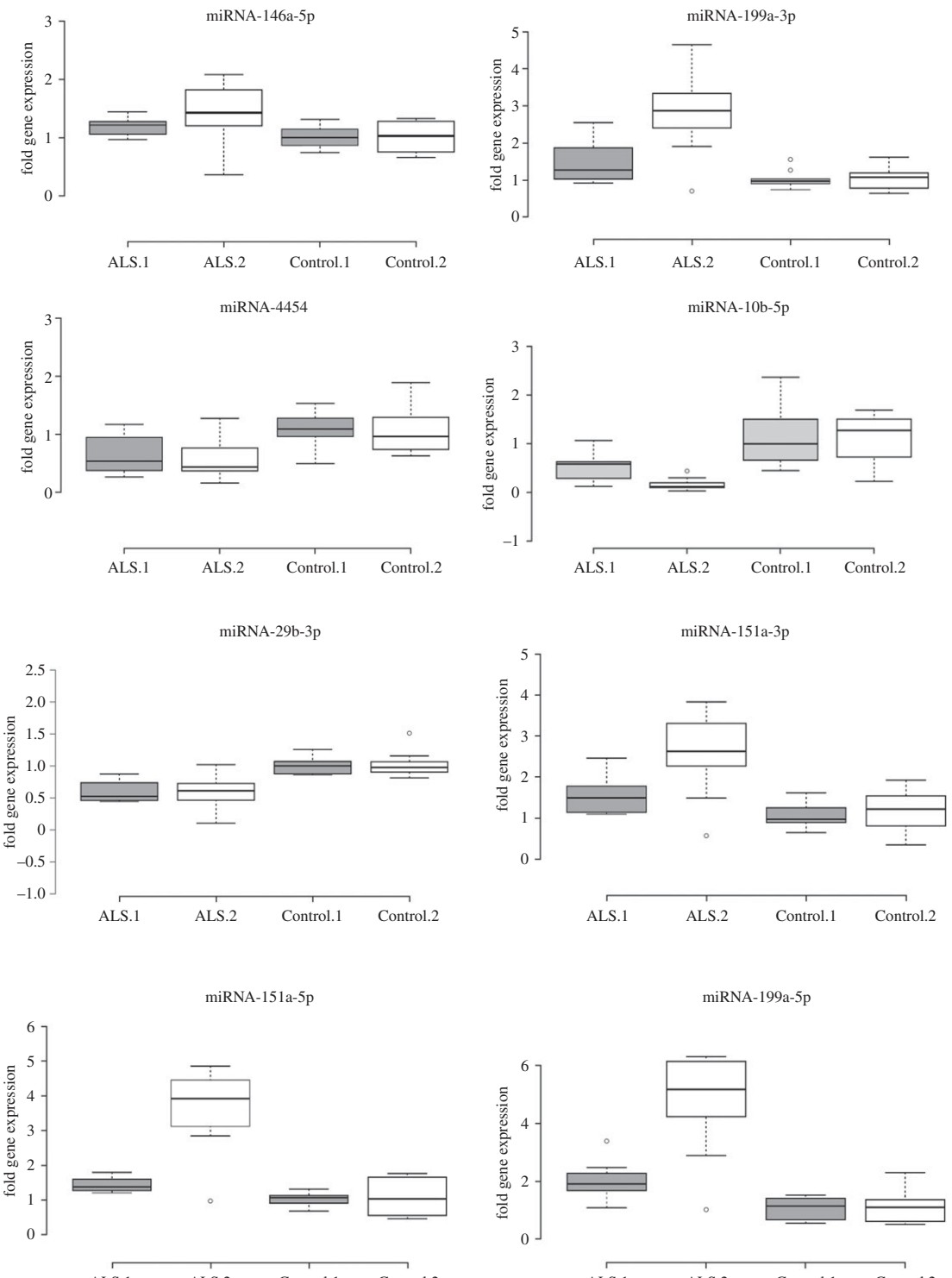

**Figure 2.** Box-plot representation of variability in gene fold expression [$2^{-(\Delta\Delta Ct)}$] in eight miRNA comparing ALS patient and healthy controls in each of two replicated experiments comprised of separate individual cohorts. All eight miRNA differed statistically between patients and controls in each of the analysed cohort experimental groups. A two-tailed Mann–Whitney $U$ Test (non-parametric based on the fact that data distribution plots did not conform to normal distributions) identified statistical differences ($p < 0.05$). ALS.1 and Control.1 (shaded box) represent the first experiment with $n = 10$ samples in each group. ALS.2 and Control.2 (open box) represent an independent replication using a new cohort of patients and controls each with $n = 10$ samples. Center lines show the medians; box limits indicate the 25th and 75th percentiles; whiskers extend 1.5 times the interquartile range from the 25th and 75th percentiles, outliers are represented by circles.

## 3.3. NGS analysis of ALS/MND patient and control plasma using NEE

Next generation sequencing analysis identified miRNA, small RNA, genome-mapped, out-mapped, high-abundance RNA and unmapped reads, the latter of which did not align to the genome. We characterized miRNA as having 18–23 nucleotide lengths. The average number of UMI-corrected reads per sample was 6.6 million, with an average percentage of mappable reads of 61.5%, indicating usable data.

## 3.4. Expression levels of miRNA

A total of 350 miRNAs were identified with a call rate ≥ 1 TPM and 219 were found to have a call rate ≥ 10 TPM. Statistical analysis of NGS data from extracellular vesicles

**Table 2.** Characterization of extracellular vesicles using nanoparticle tracking analysis suggests that vesicles are intact and parameters are consistent with exosomes. Median values reported for dominant peak.

| | neural-enriched EV | | total EV | |
| --- | --- | --- | --- | --- |
| | fluorescence | scatter | fluorescence | scatter |
| | *n* = 3 | *n* = 3 | *n* = 4 | *n* = 4 |
| particle diameter | 102 nm | 126 nm | 142 nm | 130 nm |
| full-width half-max | 95 nm | 98 nm | 101 nm | 106 nm |
| representation | 85% | 100% | 94% | 97% |
| span (90x—10x)/50x | 1.6 | 1 | 1.1 | 1.2 |

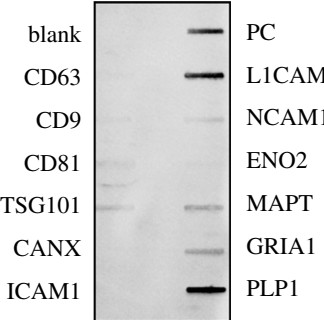

**Figure 3.** Exosome antibody detection of proteins found in neural-enriched exosome (NEE) extracts: Blank is negative control; CD63, CD9, CD81 are tetra-spanins (CD63 and CD81 were confirmed independently through ELISAs); TSG101 is a component of the ESCRT-I complex; CANX is a calnexin cell contamination marker; ICAM1 is an intercellular adhesion molecule; PC is a positive control for HRP detection; and neural marker proteins include L1 transmembrane (L1CAM), neural cell adhesion 1 (NCAM1), enolase 2 (ENO2), total tau (MAPT), glutamate receptor 1 (GRIA1) and proteolipid 1 (PLP1).

(NEE) extracted from the plasma of 10 healthy control and 10 ALS/MND patients returned 101 significantly differentially expressed miRNA ($p < 0.05$). From these 101 miRNA, 34 were chosen for relative-quantification using qPCR.

### 3.5. qPCR quantitation; miRNA QC results

We observed a steady level of expression of UniSp3 and UniSp6 in every sample indicating the RT and qPCR reactions were successful. Similar Cts for all negative controls indicate none of the samples contained inhibitors. One sample had a slightly elevated haemolysis ratio (greater than 7.0), which we determined to be insufficient to exclude from further analysis. We report Cts between 20 and 30 within all 40 samples, indicating there was enough miRNA to proceed with the downstream experiments.

### 3.6. qPCR quantitation of differentially expressed miRNA

From the 101 species of miRNA identified as differentially expressed by NGS, we selected 34 for downstream qPCR quantitation. The criteria for selection were based on (1) miRNA that were significantly differentially expressed between the controls and the patients, as determined by

NGS, and; (2) miRNA that were detected in the NGS and had previously been identified in the literature as being of interest in neurodegenerative disease.

Statistical analysis of the 34 target miRNA sequences from $n = 10$ controls and $n = 10$ patients returned highly significant comparative data, suggesting robust results. Thus, we repeated the entire experiment using identical methods but using a new cohort of patient and control samples. Statistical analysis of the replicate experiments returned eight miRNA that were differentially expressed between NEE derived from ALS/MND patients in comparison with a healthy control population of equal size in both experiment one and two (table 1). The following miRNA were analysed by qPCR but were found to be not significantly different between ALS/ MND patients and controls, or not significantly different in one of the two experiments conducted using different patient cohorts which makes them not sensitive enough for use as ALS/MND biomarkers: let-7b-5p, let-7d-3p, let-7d-5p, miR-126-3p, miR-126-5p, miR-133a-3p, miR-1-3p, miR-143-3p, miR-146a-3p, miR-194-3p, miR-23a-3p, miR-330-3p, miR-338-3p, miR-339-3p, miR-339-5p, miR-451a, miR-517a-3p, miR-584-5p, miR-625-3p, miR-708-5p and miR-744-5p.

## 4. Discussion

We identified eight miRNA sequences derived from NEE extractions that consistently and significantly differentiate ALS/MND patients from healthy controls with a single blood draw. These miRNA sequences were drawn from two experiments using different patient and control cohorts each producing the same eight miRNA sequences. We suggest that these miRNA sequences, singly or in combinations, can confirm the diagnosis of ALS/MND based on standard clinical criteria and may allow ALS/MND to be diagnosed in pre-symptomatic individuals, rapidly speeding diagnosis and treatment. Furthermore, the upregulation or downregulation of these miRNA sequences may potentially allow the effectiveness of existing or novel treatments of ALS/MND to be rapidly assessed before any clinical changes in patient disease progression or symptoms.

EVs drawn from blood plasma are important reservoirs for biomarkers for three reasons: (1) they are stable and abundant in biological fluids; (2) blood plasma is routinely drawn from patients and this procedure is relatively non-invasive when compared with lumbar punctures or tissue biopsies; and (3) the cargo of EVs contain important biomolecules including

nucleic acids and proteins. Owing to unique proteins on the surface of EVs, subpopulations of specific origins can be enriched. For example, the presence of L1CAM was used in our experiments to isolate a sub-population of neural-enriched EVs. Although L1CAM is not exclusively expressed in the brain, the enrichment process enhances the chance that the biomarkers found are related specifically to neurodegeneration. The consistency of the results across two independent experiments achieved in this report supports this method as being robust and useful in the discovery of biomarkers. We examine only miRNA in this study, but it may be possible to use the same extraction and enrichment techniques to examine proteins, lipids, and other RNA species for biomarker potential.

Katsu et al. [46], in a study with five ALS patients and five control patients using extraction methods similar to ours, identified 30 miRNA that differed between the two groups but these were not verified by qPCR. This contrasts with our study that had a larger sample size (20 per independent experiment × 2 experiments = 40 total individual samples) and was validated by qPCR. Data from our NGS experiment, which helped to identify miRNA sequences of interest for the more rigorous qPCR analysis, revealed only two overlapping miRNA sequences (miR-24-3p, miR-150-3p) with Katsu et al. [46] neither of which we chose for further study. Katsu et al. [46] suggested that the data they presented should be validated by qPCR and that larger patient cohorts are needed to determine the broader application of the identified miRNA to ALS. The majority of the miRNA found by Katsu et al. [46] were not identified in our NGS study indicating that they were either not present in the samples we examined, or that their abundance was sufficiently low as to not be recognized. Of the two miRNA that did overlap between our NGS study and the Katsu et al. [46] study, we evaluated both after the necessary statistical adjustment for false discovery rates (FDR) and found that miR-24-3p was not significant between ALS/MND patients and controls (FDR $p$-value = 0.053), but that miR-150-3p was significant between ALS/MND patients and controls (FDR $p$-value = 0.006). These two miRNA that we identified in NGS have not undergone qPCR evaluation, therefore we cannot determine the importance of these miRNA as possible ALS biomarkers. We chose not to report the other possible 67 identified miRNA from our NGS analysis until they can be further analysed in a more quantitative fashion.

Numerous miRNA have been considered as potentially valuable for ALS/MND patient biomarker investigation by other researchers using CSF, peripheral blood leucocytes, muscle tissue or plasma/serum in the absence of extracellular vesicle isolation [47–60]. It is difficult to parse the comparable value of miRNA from different biological fluids and different methods of extraction and analysis. We do note that of those miRNA identified in other studies and which were included in our list of qPCR interrogated miRNA, the following did not meet the criteria to reject a hypothesis of similar expression values between ALS/MND patients and healthy controls in our sample population: let-7b-5p, let-7d-3p, let-7d-5p, miR-126-5p, miR-133a-3p, miR-143-3p, miR-146a- 3p, miR-23a-3p, miR-338-3p, miR-451a, miR-584a-5p [49–53,56,60]. miR-146a-5p, miR-151a-5p, miR-199a-3p, miR-199a-5p were consistently significant in our analysis and found in other biofluids and fractions in other studies as important [51,60]. Of great interest is the associations of miR-151a-5p, miR-199a-3p, miR-199a-5p found by Raheja et al. [51] from circulating blood serum to be correlated with clinical ALS parameters in

a longitudinal analysis. In addition, Raheja et al. [51] found that miR-199a-5p was upregulated in both ALS and AD but that the expression levels could distinguish between ALS and AD patients. The expression of miR-146a-5p in CSF was downregulated in a sequencing study by Waller et al. [60] but they were unable to verify this result using qPCR. In our study, miR-146a-5p was analysed by qPCR and shown to be upregulated in NEE of two separate experiments using a different cohort of patients. We did not choose all the significant miRNA sequences identified in the NGS evaluation for qPCR and we are not reporting the full results of the NGS analysis because we recognize that individual miRNA should be fully evaluated using a more quantitative approach in order to understand their biological relevancy. We suggest that the extraction protocol followed in this study, extracting extracellular vesicles from blood plasma followed by enrichment of neural vesicles using L1CAM, leads to a more relevant pool of miRNA that is directly associated with neurodegenerative processes, is repeatable, and might be specifically useful as biomarkers for ALS/MND. The replication reported here using different cohorts of patients and controls supports this assertion. Ultimately, biomarkers for ALS/MND clinical diagnosis and prognosis may involve several combined approaches currently under investigation by multiple researchers. The precise disease related associations between the identified miRNA in our study and their biological targets, in the context of ALS/MND, are not yet fully known. We can draw important comparisons from studies of neurodegeneration in general.

miR-146a-5p is known to be involved in both influencing synaptic plasticity [61] and regulating the inflammatory response [62]. We report that miR-146a-5p was upregulated in the NEE of ALS/MND patient samples versus controls. Downregulation of miR-146a-5p leads to an increase in dendritic microtubule-associated protein 1B (MAP1B) translation which can reduce synaptic transmission in neurons and this pathway is part of the pathogenesis of Rett syndrome [63]. In this model, the decrease in miR-146a-5p resulted in an increase in MAP1B and corresponding AMPA receptor endocytosis. In the context of ALS/MND, upregulation of miR-146a-5p could negatively impact synaptic plasticity [61]. Similarly, miR-146a-5p may play a role in spinal muscular atrophy (SMA) as evidenced by reduced SMA astrocyte-induced motor neuron loss when miR-146a-5p is inhibited [64]. Taken together, these data suggest that an increase in miR-146a-5p, as seen in the current study, could trigger motor neuron loss which further supports the possible role of miR-146a-5p in the pathogenicity of ALS/MND.

The precise function of miR-146a-5p in ALS, however, could also be related to a role in anti-inflammation particularly within astrocytes [65–67]. Upregulation of miR-146a-5p in ALS/MND is consistent with Lu et al.'s [65] data suggesting that it binds to the 3′ UTR of TRAF6 mRNA inhibiting both mRNA and protein expression of TRAF6 resulting in a reduction of neuropathic pain. miR-146a-5p seems to be correlated with neurodegeneration in general as it has been implicated in AD where it is also upregulated in superior temporal lobe neocortex which was found to correlated with an increase in severity of disease with direct impact on immune response and inflammation [68]. We note that miR-146a has been found to be upregulated in AD brain tissues and downregulated in plasma, serum, and CSF of Alzheimer's patients [68–73]. miR-146a was not found to be upregulated in

four ALS, four PD or five schizophrenia temporal lobe neocortex tissues when compared with six control tissues [70].

Using a proprietary method of total exosome extraction (synthetic peptides, Venceremin) with a high affinity for heat shock proteins [74], Saucier *et al.* [75] found two miRNA related to ALS patients which differed from controls and which overlap with our own miRNA analysis of NEE. In the case of miR-199a-3p, they note this miRNA was downregulated in their analysis of total exosomes, while we found that it was consistently upregulated when examining the specific subpopulation of NEE. However, both our studies note miR-4454 as upregulated in ALS/MND patients when compared to control patient samples.

miR-199a is highly expressed in rat neural tissues [76]. miR199a-3p regulates the expression of mammalian target of rapamycin protein (mTOR) which plays a role in protein synthesis, cell growth, and is an important protein for axon regeneration and plasticity following central nervous system damage [77]. When upregulated, miR199a-3p should decrease the mTOR protein with a predicted negative effect on regeneration processes for damaged neurons. This miRNA sequence could be an important indicator for ALS/MND.

miR-10b-5p has been implicated in Huntington's disease pathogenicity where it was found to be upregulated in post-mortem prefrontal cortex tissue [78]. We note that miR-10b-5p was consistently downregulated in ALS/MND patients in this study. Since the presence of miR-10b-5p has been shown to suppress brain-derived neurotrophic factor (BDNF) [79], the downregulation of miR-10b-5p would be expected to increase BDNF. This result could have positive effects on memory and learning, synaptogenesis, and survival and differentiation of striatal neurons, and is consistent with an increase in BDNF found in the lymphocytes of ALS patients [80–82].

Although miR-29 sequences are well known to have anti-tumour effects, they also regulate genes related to proapoptotic/antiapoptotic pathways [83]. More research is needed to understand the role this miRNA might play in ALS/MND pathogenesis or the cellular response to disease symptoms in ALS/MND.

miR-151a-3p has previously been linked with autism and schizophrenia (miR-151), however, the fold regulation is reversed in our study of ALS/MND, being upregulated here and downregulated in these two other diseases using different sample types [84,85]. However, we note that miR-151a-3p is also upregulated in AD (noted in circulating RNA isolated directly from blood [86]) and in PD (reported from CSF exosomes [87]) but not from neural-enriched EVs as found in our study.

miR-151a-5p is thought to participate in the maintenance of cell viability through changing the cell response to oxidative stress [88]. Since the familial ALS gene mutation SOD-1 relates to an increase in mitochondrial reactive oxygen species and subsequent vulnerability to excitotoxicity [89], it is plausible that miR-151a-5p plays a role in ALS/MND.

Lastly, the upregulation miR-199a-5p, which is seen in this study, has been demonstrated to have protective effects in rat spinal cord injury models [90]. We suggest that this miRNA has similar protective effects in ALS/MND motor neuron degeneration.

# 5. Conclusion

We successfully extracted and characterized a neural-enriched subpopulation of EVs from 40 ALS/MND patient and control samples and determined that they contain sufficient miRNA to conduct NGS and qPCR. In repeated experiments using different patient and control cohorts, we have identified eight miRNA sequences that are differentially expressed in ALS/MND patients and healthy controls. This replication provides strong evidence that these miRNA sequences individually or in combination should be further investigated as ALS/MND biomarkers using larger sample sizes. Further work to compare these results with other neurodegenerative conditions is also warranted.

**Ethics.** This study was approved by the institutional ethical review board of Dartmouth Hitchcock Medical School under the FDA approved Phase IIa human clinical trial (NCT03580616) and by Innovative Research FDA Approval (3003372368). All samples were collected following written informed consent.

**Competing interests.** The Brain Chemistry Labs is filing a patent on the use of this biomarker.

**Funding.** This work was supported by Brian & Wetonnah McCoy, The Lee's Cares Foundation, The Nicholas Martin Jr. Family Foundation, The One Foundation and The Satter Foundation.

**Acknowledgements.** We thank V. Portnoy for partial Zetaview analysis, E. Stommel, Principal Investigator of the Phase IIa ALS trial at Dartmouth Medical School for ALS patient plasma samples.

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
