## [Reviewer comments · Open Biology]

Review History

RSOB-20-0055.R0 (Original submission)

Review form: Reviewer 1

Recommendation

Accept with minor revision (please list in comments)

Do you have any ethical concerns with this paper?

No

Comments to the Author

1. The authors state that biomarkers for ALS are not currently available. That is not accurate, eg Shephard et al 2017, *Neurology* 17:1137. A brief listing of other proposed biomarkers for ALS should be added.
2. Page 11, "Cts" should be spelled out.
3. The authors state that the study was replicated in two independent experiments, but they do not indicate if this involved a second cohort of patients and controls, or just a repeat extraction of NEEs from the same original cohort of 20 subjects.
4. The authors should make mention in the Discussion of additional contents of NEEs that could

be used as biomarkers in ALS, and perhaps report any such findings.

5. Were the findings in patients who were taking riluzole or endaravone different from those who were not?
6. Did the time from onset of symptoms or diagnosis have any effect on the findings?

Review form: Reviewer 2

Recommendation

Reject – article is scientifically unsound

Do you have any ethical concerns with this paper?

No

Comments to the Author

Amyotrophic Lateral Sclerosis (ALS) is a progressive fatal adult-onset neurodegenerative disorder characterized by the selective loss of lower and upper motor neurons as well as muscle degeneration. About 10% of the cases are inherited in a dominant manner (familial ALS) but most forms of ALS are sporadic (sALS) and of unknown origin. To date, there is no cure and no effective therapy. The diagnosis of ALS is mainly based on clinical assessment and electrophysiological examinations with a history of symptom. Therefore, the identification of biomarkers specific for ALS is critical for diagnostic, prognostic and therapeutic purposes. Increasing evidence points to microRNAs (miRNAs) as promising biomarkers for neurodegenerative diseases, since they are remarkably stable in human body fluids and can reflect physiological and pathological processes relevant for ALS.

Here, Banack and colleagues performed next-generation sequencing to identify miRNAs that are differentially expressed in neural-enriched extracellular vesicles extracted from human blood plasma of ALS patients (n=10) compared to non-neurological healthy controls (n=10). The authors report the identification of 101 miRNAs that are differentially expressed in ALS samples compared to controls, and further select 34 for validation using quantitative PCR. By repeating the whole experiment a second time with the same approach, a total of eight miRNAs is ultimately proposed to be specifically associated with ALS disease, which may help for early disease diagnosis.

The effort here identifies eight miRNAs whose levels are increased or decreased in plasma from ALS patients by a fold change range of 1.2 - 4.2 and -1.7 - -7.0, respectively. These findings could be of interest as these miRNAs could represent potential diagnostic biomarkers for ALS. However, in its present format the manuscript does not convincingly demonstrate that it is the case. Critical additional work is needed to ascertain the relevance of the results reported here. Here are the concerns:

1. This work adds to the numerous previous studies reporting ALS-specific miRNAs as potential biomarkers, including the analyses of ALS plasma samples by Takahashi et al (Mol Brain, 2015) or more recently by Katsu et al, (Neurosci. Lett, 2019) who also identified miRNAs from neural-enriched extracellular vesicles from ALS plasma samples). Disappointingly, none of these previous studies are cited by Banack et al. Comparison of the current findings with previous efforts would be valuable and could provide key insights in the elucidation of the most relevant miRNAs. In view of this reviewer, this is critical.

2. The key data, i.e. fold change levels of the eight miRNAs is presented in a summary table where only the median value for control and ALS patients is provided. No standard deviation values between patients is included to enable defining variability between patients. This is essential. A graph with the individual measures obtained per patient should be included for each

miRNA identified.

3. Efforts should be made to improve the writing/description of the results section. In its present format, this section is very technical and reads almost as a material and method section. It is very difficult to capture the key findings, and significantly reduces the interest of the reader.

4. The quality of figure 1 and its description is poor. It is not possible to ascertain the purity/enrichment for the neural- enriched extracellular vesicles.

In sum, the manuscript in its present format is not recommended for publication and the concerns raised here should be addressed.

Additional concerns:

1. There is no information about the genetics of the ALS patients used in the study. Have the patients been screened for the most recent genetic forms? This information is particularly important in light of the comparison with the previous efforts.

2. It would be of interest to perform a longitudinal analysis to assess the levels of the identified miRNAs during disease progression.

3. As the authors acknowledge, it would be valuable to determine whether these miRNAs are ALS specific or also present in other neurological instances.

Decision letter (RSOB-20-0055.R0)

14-Apr-2020

Dear Dr Cox,

We are writing to inform you that your manuscript RSOB-20-0055 entitled "Identification of an miRNA fingerprint from neural-enriched extracellular vesicles from plasma of amyotrophic lateral sclerosis patients" has, in its current form, been rejected for publication in Open Biology.

This action has been taken on the advice of the referees, who have recommended that substantial revisions are necessary. With this in mind we are willing to consider a resubmission, provided all the comments of the referees are taken into account. Please note this is not a provisional acceptance.

The resubmission will be treated as a new manuscript and will re-enter the review process. Every attempt will be made to use the original referees, but this cannot be guaranteed. Please note that resubmissions must be submitted within six months of the date of this email. In exceptional circumstances, extensions may be possible if agreed with the Editorial Office. Manuscripts submitted after this date will be automatically rejected.

Please find below the comments made by the referees, not including confidential reports to the Editor, which I hope you will find useful. If you do choose to resubmit your manuscript, please upload a 'response to referees' document including details of how you have responded to the comments, and the adjustments you have made.

To upload a resubmitted manuscript, log into <http://mc.manuscriptcentral.com/rsob> and enter your Author Centre, where you will find your manuscript title listed under "Manuscripts with Decisions." Under "Actions," click on "Create a Resubmission." Please be sure to indicate in your cover letter that it is a resubmission, and supply the previous reference number.

Sincerely,
The Open Biology Team
mailto: openbiology@royalsociety.org

Board Member's comments to Author(s):

Due to the justified concerns of one of the two reviewers, we are sorry to let you know that we need to reject the manuscript in its present form.

Reviewer(s)' Comments to Author(s):

Referee: 1

Comments to the Author(s)

1. The authors state that biomarkers for ALS are not currently available. That is not accurate, eg Shephard et al 2017, *Neurology* 17:1137. A brief listing of other proposed biomarkers for ALS should be added.
2. Page 11, "Cts" should be spelled out.
3. The authors state that the study was replicated in two independent experiments, but they do not indicate if this involved a second cohort of patients and controls, or just a repeat extraction of NEEs from the same original cohort of 20 subjects.
4. The authors should make mention in the Discussion of additional contents of NEEs that could be used as biomarkers in ALS, and perhaps report any such findings.
5. Were the findings in patients who were taking riluzole or endaravone different from those who were not?
6. Did the time from onset of symptoms or diagnosis have any effect on the findings?

Referee: 2

Comments to the Author(s)

Amyotrophic Lateral Sclerosis (ALS) is a progressive fatal adult-onset neurodegenerative disorder characterized by the selective loss of lower and upper motor neurons as well as muscle degeneration. About 10% of the cases are inherited in a dominant manner (familial ALS) but most forms of ALS are sporadic (sALS) and of unknown origin. To date, there is no cure and no effective therapy. The diagnosis of ALS is mainly based on clinical assessment and electrophysiological examinations with a history of symptom. Therefore, the identification of biomarkers specific for ALS is critical for diagnostic, prognostic and therapeutic purposes. Increasing evidence points to microRNAs (miRNAs) as promising biomarkers for neurodegenerative diseases, since they are remarkably stable in human body fluids and can reflect physiological and pathological processes relevant for ALS.

Here, Banack and colleagues performed next-generation sequencing to identify miRNAs that are differentially expressed in neural-enriched extracellular vesicles extracted from human blood plasma of ALS patients (n=10) compared to non-neurological healthy controls (n=10). The authors report the identification of 101 miRNAs that are differentially expressed in ALS samples compared to controls, and further select 34 for validation using quantitative PCR. By repeating the whole experiment a second time with the same approach, a total of eight miRNAs is ultimately proposed to be specifically associated with ALS disease, which may help for early disease diagnosis.

The effort here identifies eight miRNAs whose levels are increased or decreased in plasma from ALS patients by a fold change range of 1.2 - 4.2 and -1.7 - -7.0, respectively. These findings could be of interest as these miRNAs could represent potential diagnostic biomarkers for ALS. However, in its present format the manuscript does not convincingly demonstrate that it is the case. Critical additional work is needed to ascertain the relevance of the results reported here. Here are the concerns:

1. This work adds to the numerous previous studies reporting ALS-specific miRNAs as potential biomarkers, including the analyses of ALS plasma samples by Takahashi et al (Mol Brain, 2015) or more recently by Katsu et al, (Neurosci. Lett, 2019) who also identified miRNAs from neural-enriched extracellular vesicles from ALS plasma samples). Disappointingly, none of these previous studies are cited by Banack et al. Comparison of the current findings with previous efforts would be valuable and could provide key insights in the elucidation of the most relevant miRNAs. In view of this reviewer, this is critical.
2. The key data, i.e. fold change levels of the eight miRNAs is presented in a summary table where only the median value for control and ALS patients is provided. No standard deviation values between patients is included to enable defining variability between patients. This is essential. A graph with the individual measures obtained per patient should be included for each miRNA identified.
3. Efforts should be made to improve the writing/description of the results section. In its present format, this section is very technical and reads almost as a material and method section. It is very difficult to capture the key findings, and significantly reduces the interest of the reader.
4. The quality of figure 1 and its description is poor. It is not possible to ascertain the purity/enrichment for the neural- enriched extracellular vesicles.

In sum, the manuscript in its present format is not recommended for publication and the concerns raised here should be addressed.

Additional concerns:

1. There is no information about the genetics of the ALS patients used in the study. Have the patients been screened for the most recent genetic forms? This information is particularly important in light of the comparison with the previous efforts.
2. It would be of interest to perform a longitudinal analysis to assess the levels of the identified miRNAs during disease progression.
3. As the authors acknowledge, it would be valuable to determine whether these miRNAs are ALS specific or also present in other neurological instances.

Author's Response to Decision Letter for (RSOB-20-0055.R0)

See Appendix A.

RSOB-20-0116.R0

Review form: Reviewer 1

Recommendation

Accept as is

Do you have any ethical concerns with this paper?

No

Comments to the Author

Thank you for responding to my comments.

Review form: Reviewer 2

Recommendation

Accept with minor revision (please list in comments)

Do you have any ethical concerns with this paper?

No

Comments to the Author

Banack and colleagues have overall improved their manuscript, providing clarification on aspects of the experimental design, results and data analysis that were previously unclear. Figure 2 is an important addition to inform the reader about the variability in the levels measured between patients for each of the 8 miRNAs that the authors focused on.

The comparison of the present findings with previously published reports now included in the discussion is informative.

Lastly, it is not clear whether the authors made/will make available the raw sequencing data to the Gene Expression Omnibus (GEO) database for example, as it is typically done when RNA sequencing data is published. Furthermore, a table listing all the miRNAs identified by sequencing (with their respective fold change and statistics- like the one provided for the Q-PCR data-Table 1) would be of value for the scientific community.

Decision letter (RSOB-20-0116.R0)

20-May-2020

Dear Dr Cox,

We are pleased to inform you that your manuscript RSOB-20-0116 entitled "Identification of an miRNA fingerprint from neural-enriched extracellular vesicles from plasma of amyotrophic lateral sclerosis patients" has been accepted by the Editor for publication in Open Biology. The reviewer(s) have recommended publication, but also suggest some minor revisions to your manuscript. Therefore, we invite you to respond to the reviewer(s)' comments and make sure your share your supporting data available.

Please submit the revised version of your manuscript within 7 days. If you do not think you will be able to meet this date please let us know immediately and we can extend this deadline for you.

- 1) A text file of the manuscript (doc, txt, rtf or tex), including the references, tables (including captions) and figure captions. Please remove any tracked changes from the text before submission. PDF files are not an accepted format for the "Main Document".
- 2) A separate electronic file of each figure (tiff, EPS or print-quality PDF preferred). The format should be produced directly from original creation package, or original software format. Please note that PowerPoint files are not accepted.
- 3) Electronic supplementary material: this should be contained in a separate file from the main text and meet our ESM criteria (see <http://royalsocietypublishing.org/instructions-authors#question5>). All supplementary materials accompanying an accepted article will be treated as in their final form. They will be published alongside the paper on the journal website and posted on the online figshare repository. Files on figshare will be made available approximately one week before the accompanying article so that the supplementary material can be attributed a unique DOI.

Online supplementary material will also carry the title and description provided during submission, so please ensure these are accurate and informative. Note that the Royal Society will not edit or typeset supplementary material and it will be hosted as provided. Please ensure that the supplementary material includes the paper details (authors, title, journal name, article DOI). Your article DOI will be 10.1098/rsob.2016[last 4 digits of e.g. 10.1098/rsob.20160049].

- 4) A media summary: a short non-technical summary (up to 100 words) of the key findings/importance of your manuscript. Please try to write in simple English, avoid jargon, explain the importance of the topic, outline the main implications and describe why this topic is newsworthy.

Images

Data-Sharing

It is a condition of publication that data supporting your paper are made available. Data should be made available either in the electronic supplementary material or through an appropriate

repository. Details of how to access data should be included in your paper. Please see <http://royalsocietypublishing.org/site/authors/policy.xhtml#question6> for more details.

Data accessibility section

Sincerely,
The Open Biology Team
<mailto:openbiology@royalsociety.org>

Reviewer(s)' Comments to Author:

Referee: 1

Comments to the Author(s)
Thank you for responding to my comments.

Referee: 2

Comments to the Author(s)

Banack and colleagues have overall improved their manuscript, providing clarification on aspects of the experimental design, results and data analysis that were previously unclear. Figure 2 is an important addition to inform the reader about the variability in the levels measured between patients for each of the 8 miRNAs that the authors focused on.

The comparison of the present findings with previously published reports now included in the discussion is informative.

Lastly, it is not clear whether the authors made/will make available the raw sequencing data to the Gene Expression Omnibus (GEO) database for example, as it is typically done when RNA sequencing data is published. Furthermore, a table listing all the miRNAs identified by sequencing (with their respective fold change and statistics- like the one provided for the Q-PCR data-Table 1) would be of value for the scientific community.

Author's Response to Decision Letter for (RSOB-20-0116.R0)

See Appendix B.

Decision letter (RSOB-20-0116.R1)

22-May-2020

Dear Dr Cox

We are pleased to inform you that your manuscript entitled "An miRNA fingerprint using neural-enriched extracellular vesicles from blood plasma: toward a biomarker for ALS/MND" has been accepted by the Editor for publication in Open Biology.

Article processing charge

Please note that the article processing charge is immediately payable. A separate email will be sent out shortly to confirm the charge due. The preferred payment method is by credit card; however, other payment options are available.

Sincerely,

The Open Biology Team

mailto: openbiology@royalsociety.org

Appendix A

Response to Referees (RSOB-20-0055):

We thank the reviewers for the comments to improve this manuscript for publication. We are grateful for the careful analysis offered by the referees and feel that it is a better paper as a result of their comments. We have addressed every comment with a summary of the changes listed below each comment.

Referee: 1

Comments to the Author(s)

1. The authors state that biomarkers for ALS are not currently available. That is not accurate, eg Shephard et al 2017, Neurology 17:1137. A brief listing of other proposed biomarkers for ALS should be added.

We thank this review for pointing out this discrepancy. We note that many ALS biomarkers have been proposed but that these are not yet being used for diagnosis in the clinical setting. We have amended this statement in the abstract which now reads: “Biomarkers for amyotrophic lateral sclerosis (ALS) are currently not clinically available for disease diagnosis or analysis of disease progression.”

In addition, we have added the following paragraph to more fully explain this concept in the Introduction: “To date, an ALS diagnosis is based on clinical features with the elimination of alternative diagnoses and supporting data retrieved from electromyograms, nerve conduction studies, muscle biopsies, magnetic field imaging, and biofluid analysis (Pasinetti et al. 2006; Mayo clinic 2020). The search for ALS biomarkers useful for diagnosis, prognosis, and the efficacy of drug therapy includes a variety of molecules found in biofluids and other techniques including: heavy and light chain neurofilaments, TAR DNA-binding protein 43 (TDP-43), a lipid peroxidation product (4-hydroxy-2,3-nonanal), a urinary neurotrophin receptor p75 extracellular domain, cystatin C, mRNA, miRNA, extracellular glutamate, markers of inflammation, immune, and glial activation, electrical impedance myography, rate of disease progression, spinal cord imaging, and others (Ferrarese et al. 2001; Simpson et al. 2004; Boyland et al. 2009; Wilson et al. 2010; Bede et al. 2012; Rutkove et al. 2012; Tarasiuk et al. 2012; Feneberg et al. 2014; Lu et al. 2015a; 2015b; Labra et al. 2016; Shephard et al. 2017; Poesen et al. 2019; Hosaka et al. 2019). Thus far, none of these biomarkers have been sufficiently validated to be incorporated into the clinical standard of care (Pasinetti et al. 2006; von Neuhoff et al. 2012; Tarasiuk et al. 2012; Hosaka et al. 2019; Poesen et al. 2019). ”

2. Page 11, "Cts" should be spelled out.

The following sentences were added to define what a Ct is. “In a real time PCR assay a positive reaction is detected by accumulation of a fluorescent signal. The cycle threshold (Ct) is defined as the number of cycles required for the fluorescent signal to cross the threshold (ie exceed background levels).” In addition, Ct has been added to the abbreviation list.

3. The authors state that the study was replicated in two independent experiments, but they do not indicate if this involved a second cohort of patients and controls, or just a repeat extraction of NEEs from the same original cohort of 20 subjects.

The original experiment was conducted with 10 ALS and 10 controls. The replication was a separate experiment done with 10 different ALS patients and 10 different controls for a total analysis of 40 individuals. This has been rewritten throughout the manuscript to clarify this point and we thank the reviewer for pointing out that this was not clear in the earlier version.

4. The authors should make mention in the Discussion of additional contents of NEEs that could be used as biomarkers in ALS, and perhaps report any such findings.

The analysis of other NEE cargo is still on-going and, should the results be promising, will be included in a subsequent publication. The purpose of the current manuscript is to report only the results of the miRNA analyses which we feel makes for a more focused paper of these important findings.

We have added the following sentences to clarify the potential to find other NEE cargo using these same methods. “The consistency of the results across two independent experiments achieved in this report supports this method as being robust and useful in the discovery of biomarkers. We examine only miRNA in this paper but it may be possible to use the same extraction and enrichment techniques to examine proteins, lipids, and other RNA species for biomarker potential.”

5. Were the findings in patients who were taking riluzole or endaravone different from those who were not?

The data presented here were analyzed within a blind to all other factors, due to small sample cohorts. Since the results were cohesive and statistically significant by diagnosis alone in two separate cohorts analyzed, we did not dilute the sample size by separating out other factors. The statistical analysis was conducted using non-parametric statistics which are more conservative than parametric statistics. When larger sample sizes become available to repeat these experiments, it would then be appropriate to perform multi-variate analyses and include additional factors such as time since disease onset and drug interactions that could influence miRNA concentrations.

6. Did the time from onset of symptoms or diagnosis have any effect on the findings?

See answer to comment #5 above.

Referee: 2

Comments to the Author(s)

Amyotrophic Lateral Sclerosis (ALS) is a progressive fatal adult-onset neurodegenerative disorder characterized by the selective loss of lower and upper motor neurons as well as muscle degeneration. About 10% of the cases are inherited in a dominant manner (familial ALS) but most forms of ALS are sporadic (sALS) and of unknown origin. To date, there is no cure and no effective therapy. The diagnosis of ALS is mainly based on clinical assessment and electrophysiological examinations with a history of symptom. Therefore, the identification of biomarkers specific for ALS is critical for diagnostic, prognostic and therapeutic purposes. Increasing evidence points to microRNAs (miRNAs) as promising biomarkers for neurodegenerative diseases, since they are remarkably stable in human body fluids and can reflect physiological and pathological processes relevant for ALS.

Here, Banack and colleagues performed next-generation sequencing to identify miRNAs that are differentially expressed in neural-enriched extracellular vesicles extracted from human blood plasma of ALS patients (n=10) compared to non-neurological healthy controls (n=10). The authors report the identification of 101 miRNAs that are differentially expressed in ALS samples compared to controls, and further select 34 for validation using quantitative PCR. By repeating the whole experiment a second time with the same approach, a total of eight miRNAs is ultimately proposed to be specifically associated with ALS disease, which may help for early disease diagnosis.

The effort here identifies eight miRNAs whose levels are increased or decreased in plasma from ALS patients by a fold change range of 1.2 - 4.2 and -1.7 - -7.0, respectively. These findings could be of interest as these miRNAs could represent potential diagnostic biomarkers for ALS. However, in its present format the manuscript does not convincingly demonstrate that it is the case. Critical additional work is needed to ascertain the relevance of the results reported here. Here are the concerns:

1. This work adds to the numerous previous studies reporting ALS-specific miRNAs as potential biomarkers, including the analyses of ALS plasma samples by Takahashi et al (Mol Brain, 2015) or more recently by Katsu et al, (Neurosci. Lett, 2019) who also identified miRNAs from neural-enriched extracellular vesicles from ALS plasma samples). Disappointingly, none of these previous studies are cited by Banack et al. Comparison of the current findings with previous efforts would be valuable and could provide key insights in the elucidation of the most relevant miRNAs. In view of this reviewer, this is critical.

The following two paragraphs have been added to the discussion: “In a study with five ALS patients and five control patients using extraction methods similar to ours Katsu et al. (2019) identified 30 miRNA that differed between the two groups but these were not verified by qPCR. We contrast this with our study that had a larger sample size (20 per independent experiment x 2 experiments = 40 total individual samples) and was validated by qPCR. Data from our NGS experiment, which helped to identify miRNA of interest for the more rigorous qPCR analysis, revealed only two overlapping miRNA (hsa-miR-24-3p, hsa-miR-150-3p) with Katsu et al. (2019) neither of which we chose for

further study. As acknowledged by Katsu et al. (2019), the data they presented requires validation by RT-PCR and larger patient cohorts are needed to evaluate the broad application of the identified miRNA to ALS. The majority of the miRNA found by Katsu et al. (2019) were not identified in our NGS study indicating that they were either not present in the samples we examined, or that their abundance was sufficiently low as to not be recognized. This suggests that they may not be useful for future studies in ALS patients as general markers. Of the two miRNA that did overlap between our NGS study and the equivalent Katsu et al. (2019) study, we evaluated both after the necessary statistical adjustment for false discovery rates (FDR) and found that miR-24-3p was not significant between ALS patients and controls (FDR p-value = 0.053), but that hsa-miR-150-3p was significant between ALS patients and controls (FDR p-value = 0.006). Since this miRNA has not undergone proper qPCR evaluation, we cannot determine the importance of this comparison to the study of ALS biomarkers and have chosen to not report the other possible 67 identified miRNA from our NGS analysis until they can be further analyzed in a more quantitative fashion.

Numerous miRNA have been investigated as potentially valuable for ALS patient biomarker investigation by other researchers using cerebrospinal fluid, peripheral blood leukocytes, muscle tissue, or plasma/ serum in the absence of extracellular vesicle isolation (Butovsky et al. 2012; de Felice et al. 2012; de Felice et al. 2014; Toivonen et al. 2014; Takahashi et al. 2015; Chen et al. 2016; de Andrade et al. 2016; Tasca et al. 2016; Sheinerman et al. 2017; Waller et al. 2017a; Waller et al. 2017b; Liguori et al. 2018; Raheja et al. 2018; Vrabec et al. 2018). It is difficult to parse the comparable value of miRNA using different biological fluids and different methods of extraction and analysis. We do note that of those miRNA identified in other studies and which were included in our list of qPCR interrogated miRNA, the following did not meet the criteria to reject a hypothesis of similar expression values between ALS patients and healthy controls: let-7b-5p, let-7d-3p, let-7d-5p, miR-126-5p, miR-133a-3p, miR-143-3p, miR-146a-3p, miR-23a-3p, miR-338-3p, miR-451a, miR-584a-5p (deFelice et al. 2014; Chen et al. 2016; Waller et al. 2017b; Waller et al. 2018; Liguori et al. 2018; Raheja et al. 2018; Vrabec et al. 2018). miR-146a-5p, miR-151a-5p, miR-199a-3p, miR-199a-5p were consistently significant in our analysis and found in other biofluids and fractions in other studies as important (Raheja et al. 2018; Waller et al. 2018). Of great interest is the associations of miR-151a-5p, miR-199a-3p, miR-199a-5p found by Raheja et al. (2018) from circulating blood serum to be correlated with clinical ALS parameters in a longitudinal analysis. In addition, Raheja et al. (2018) found that miR-199a-5p was upregulated in both ALS and AD but that the expression levels could distinguish between ALS and AD patients. The expression of miR-146a-5p in CSF was down-regulated in a sequencing study by Waller et al. (2018) but they were unable to verify this results using qPCR. In this study, miR-146a-5p was analyzed by qPCR and shown to be up-regulated in NEE of two separate experiments using a different cohort of patients. We did not choose all the significant miRNA identified in the NGS evaluation for qPCR and we are not reporting the full results of the NGS analysis because we recognize that individual miRNA should be fully evaluated using a more quantitative approach in order to understand the biological relevancy. We suggest that the extraction protocol followed in this study, extracting extracellular vesicles from blood plasma followed by enrichment of neural vesicles using L1CAM, leads to a more relevant pool of miRNA that is directly associated with neurodegenerative processes, repeatable, and might be specifically useful as ALS biomarkers. The replication reported here using different cohorts of patients and controls supports this assertion. Ultimately, biomarkers for ALS

disease clinical diagnosis and prognosis may involve several combined approaches currently under investigation by multiple researchers. The precise disease related associations between the identified miRNA in this study and their biological targets, in the context of ALS, are not yet fully known. We can draw important comparisons from studies of neurodegeneration in general. ”

2. The key data, i.e. fold change levels of the eight miRNAs is presented in a summary table where only the median value for control and ALS patients is provided. No standard deviation values between patients is included to enable defining variability between patients. This is essential. A graph with the individual measures obtained per patient should be included for each miRNA identified.

Due the small sample sizes and the absence of data normality, it is appropriate to report the central tendency in terms of medians rather than using a mean. Medians do not have a standard deviation and therefore the variability cannot be presented in this way. We note that our statistical analysis was conducted using non-parametric statistics which is a more conservative approach in identifying comparative differences. However, we acknowledge the validity of the comment by this reviewer in wanting to have information on patient variability. As a result, we have added a box-plot graph as a way to illustrate this variability.

3. Efforts should be made to improve the writing/description of the results section. In its present format, this section is very technical and reads almost as a material and method section. It is very difficult to capture the key findings, and significantly reduces the interest of the reader.

We appreciate this comment and have rewritten the Results section accordingly.

4. The quality of figure 1 and its description is poor. It is not possible to ascertain the purity/enrichment for the neural- enriched extracellular vesicles.

Figure one has been eliminated and the data reported in a new Table 1 and in the body of the text.

Additional concerns:

1. There is no information about the genetics of the ALS patients used in the study. Have the patients been screened for the most recent genetic forms? This information is particularly important in light of the comparison with the previous efforts.

Genetic analysis was not performed on these patients as it was outside the scope of this study.

2. It would be of interest to perform a longitudinal analysis to assess the levels of the identified miRNAs during disease progression.

A longitudinal analysis is being considered for publication at a later time.

3. As the authors acknowledge, it would be valuable to determine whether these miRNAs are ALS specific or also present in other neurological instances.

Analysis of miRNA from NEE using patients with other neurological diseases is on-going and will be the focus of future experiments and publications. This comparative analysis is outside the scope of this manuscript, but we thank the reviewer for validation of these next steps.

Appendix B

Response to referees

Referee 1:

Thank you for your helpful editorial suggestions.

Referee 2:

We are grateful for the comments which have improved the quality of this manuscript. Target sequences used for identification of the eight miRNA have been included in electronic supplementary material. We are still mining the NGS data and performing qPCR experiments on additional sequences, which will be incorporated into future publications.